# A Real Application of an Autonomous Industrial Mobile Manipulator within Industrial Context

Jose Luis Outón [1,2,*] , Ibon Merino [1,2] , Iván Villaverde [1] , Aitor Ibarguren [1] , Héctor Herrero [1] , Paul Daelman [1] and Basilio Sierra [2]

1 Tecnalia Research and Innovation, Basque Research and Technology Alliance (BRTA), Industry and Transport Division, 20009 San Sebastián, Spain; ibon.merino@tecnalia.com (I.M.); ivan.villaverde@tecnalia.com (I.V.); aitor.ibarguren@tecnalia.com (A.I.); hector.herrero@tecnalia.com (H.H.); paul.daelman@tecnalia.com (P.D.)
2 Robotics and Autonomous Systems Group, University of the Basque Country UPV/EHU, 20009 San Sebastián, Spain; b.sierra@ehu.eus
* Correspondence: joseluis.outon@tecnalia.com

**Abstract:** In modern industry there are still a large number of low added-value processes that can be automated or semi-automated with safe cooperation between robot and human operators. The European SHERLOCK project aims to integrate an autonomous industrial mobile manipulator (AIMM) to perform cooperative tasks between a robot and a human. To be able to do this, AIMMs need to have a variety of advanced cognitive skills like autonomous navigation, smart perception and task management. In this paper, we report the project's tackle in a paradigmatic industrial application combining accurate autonomous navigation with deep learning-based 3D perception for pose estimation to locate and manipulate different industrial objects in an unstructured environment. The proposed method presents a combination of different technologies fused in an AIMM that achieve the proposed objective with a success rate of 83.33% in tests carried out in a real environment.

**Keywords:** autonomous industrial mobile manipulator; deep learning; robotics; perception; sensor fusion; autonomous navigation; computer vision; skills; state machine

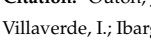

## 1. Introduction

Industrial processes have undergone a number of transformations and improvements, from the early manufacturing processes of the previous centuries to the present day, achieving high levels of efficiency and automation. As Nye [1] describes in his work on the comparative review of 100 years of industrial evolution, today's assembly line "in a new way is more productive than ever", thanks to robotics.

Nevertheless, many low value and repetitive processes are yet to be automated [2]. This lack of automation in certain tasks, usually performed by low-skilled operators, has been caused mainly by the cost of replacing low-wage operators with expensive automation systems that require very invasive changes in the area and a long reprogramming time by qualified specialists. In addition, replacing low-skilled operators with high-skilled ones is difficult and expensive. Newer education paradigms [3] have focused on the training of these high-skilled operators. However, the fast technology development and an aging workforce makes it difficult to maintain a proper high-skilled crew. This requires new solutions to assist operators and provide collaborative work environments [4]. It is expected that this successful transformation could ensure that robotics and autonomous transportation are able to create a global job growth of 1.36% [5].

The rise of autonomous systems and robotics, especially collaborative robots, is opening new market possibilities. A new trend for flexible and collaborative robotics is spreading in the industry in the form of autonomous industrial mobile manipulators (AIMM) [6]. These hybrid systems combine two widespread and mature technologies:

manipulation by robotic arms and mobile robotics. These capabilities are merged and improved with innovative techniques of artificial perception so that the robot has greater autonomy and decision-making capabilities.

These types of autonomous systems are fitted to the new industrial paradigm that has been in development in recent years: the fourth industrial revolution or "Industry 4.0". This term became publicly known in 2011, when an association of representatives from the business, political and academic world promoted the idea as an approach to strengthen the competitiveness of the German manufacturing industry [7]. In this paradigm, manufacturing and logistics processes take the form of cyber–physical production systems (CPPS) which make use of information and communications networks to interchange large amounts of information. Since its emergence, the concept of Industry 4.0 has attracted lots of attention and research [8] and numerous academic publications, practical articles and conferences have discussed this topic [9]. Similarly, the industrial robots manufacturing industry has adapted to this new approach, and in Europe alone the number of advanced robots developed almost doubled between 2004 and 2016 [10]. Among these new advanced robots, we can classify the previously mentioned AIMMs that are in the center of this report.

In a dynamic manufacturing environment, both the working areas and the manufacturing process itself are constantly changing. Thus, autonomous systems must be robust to changes. The environment perception and the capacity of readjustment must be fast and efficient.

The purpose of the work reported here is to provide a novel solution to one of the challenges proposed in the "co-manipulation of large aeronautic parts by dual-arm mobile manipulator" use case of the European project SHERLOCK [11]. This project aims to introduce the latest safe robotic technologies, including collaborative arms with high payload, exoskeletons and mobile manipulators in various production environments, enhancing them with intelligent mechatronics and AI-based cognition. The result will be creating efficient human–robot collaboration (HRC) stations that are designed to be safe and ensure the acceptance and well being of operators. Another fundamental pillar of the SHERLOCK project is the transition from traditional robotics to the new Industry 4.0 paradigm, where robotics are more flexible, autonomous, collaborative and interconnected. Within the project, other works that support it have already been published [12–14].

The scientific and technological objectives of the SHERLOCK project are summarized as follows:

- Bring recent research developments to a real world application;
- Development of a soft robotics collaborative production station;
- Novel human-centered interaction, collaboration and awareness;
- Artificial intelligence enabled cognition for autonomous human–robot collaborative applications;
- SHERLOCK modules for the design and certification of safe HRC applications.

In this context, the work reported here focuses on the second and fourth objectives, aiming to solve the challenge of recognition and detection of the supports of long aeronautical pieces by means of artificial vision techniques and deep learning. This will enable the AIMM to be able to locate the part and grasp it safely for the task that is entrusted to it. For those objectives, a lot of emphasis will be put on bringing recent methods and techniques found in the literature to the real world, along with using the currently well established, state-of-the-art methods.

The paper is distributed as follows. In Section 1.1, the objectives to be achieved in this work are described. Section 2 introduces the global system, beginning with the presentation of the AIMM and its technical details, Section 2.1, followed by the explanation of the navigation techniques used Section 2.1.1. Section 2.2 describes the perception system used to detect the required objects. The depth learning method used is explained in Sections 2.2.1 and 2.2.2. The calibration methods used for both the robot camera and the gripper TCP are described in Section 2.3. The control of the process is presented in Section 2.4 which describes the programming based on skills and the control and management of these skills by means of

a state machine. Finally, the experiment is defined in Section 3 and in Section 4 the results obtained in the tests carried out are displayed. The paper ends with the discussion and future work in.

### 1.1. Objectives

The objective of this work was for the robot to be able to autonomously carry out a sequence of tasks using a combination of different state-of-the-art technologies in a real work environment. This sequence of tasks represent a prototypical application in industry that AIMMs should be able to accomplish. To successfully carry out this task, the robot should possess the following skills:

- Autonomous navigation between different objectives;
- Precise navigation to ensure that the target position is good enough;
- Be able to perceive the environment and make decisions to adapt to it;
- Ensure that the task is carried out safely in a collaborative environment;
- A computer vision system able to detect and identify the desired object using deep learning techniques;
- Be able grasp or touch the centroid of the recognized object;
- High-level management of the tasks to be carried out as well as error management.

The advances made in the implementation of such skills in the SHERLOCK project AIMM will be presented in this report. The main contributions of this work are the successful implementation of the mentioned task, integrating and tuning different available techniques and methods in one system and bringing them to a real world application. In addition, a new RGB-D training dataset, using synthetic images from real industrial parts, has been compiled and will be made available.

## 2. System Description

### 2.1. Innovative AIMM

The robotic system used in this paper is shown in Figure 1. It is an innovative robot designed by the authors and presented in [15], where it was described extensively. As this system meets all the criteria established in [16], it can be regarded as an AIMM. As such, it consists of two main parts: a drive system and a manipulation system.

For the drive system, the kinematic configuration is based on the Swerve Drive approach [17], consisting of four motor wheels driven in translation and rotation by eight motors. While not truly holonomic, the Swerve Drive's omnidirectional capability gives the robot great versatility. The medium-sized wheels offer good stability and the capacity to overcome small obstacles and ground irregularities. The AIMM reaches a maximum speed of 3 m/s, although it is limited to 2 m/s by software for safety reasons.

For the manipulation system, current needs of the industry show that processes require to perform complex object manipulation tasks. To provide the robot with greater versatility and improved manipulation capabilities, it has been equipped with two Universal Robots UR10 robotic arms. In this way, not only the total payload is increased, but it also allows more complex manipulation strategies by means of coordinated movements of the two arms [18].

The arms are mounted on a rotating vertical axis at the front, providing two additional degrees of freedom to the robotic arms (670 mm of elevation and $\pm 350^{\circ}$ of rotation). This way the AIMM greatly increases its reachability (up to 2.5 m high) and the volume of the working space.

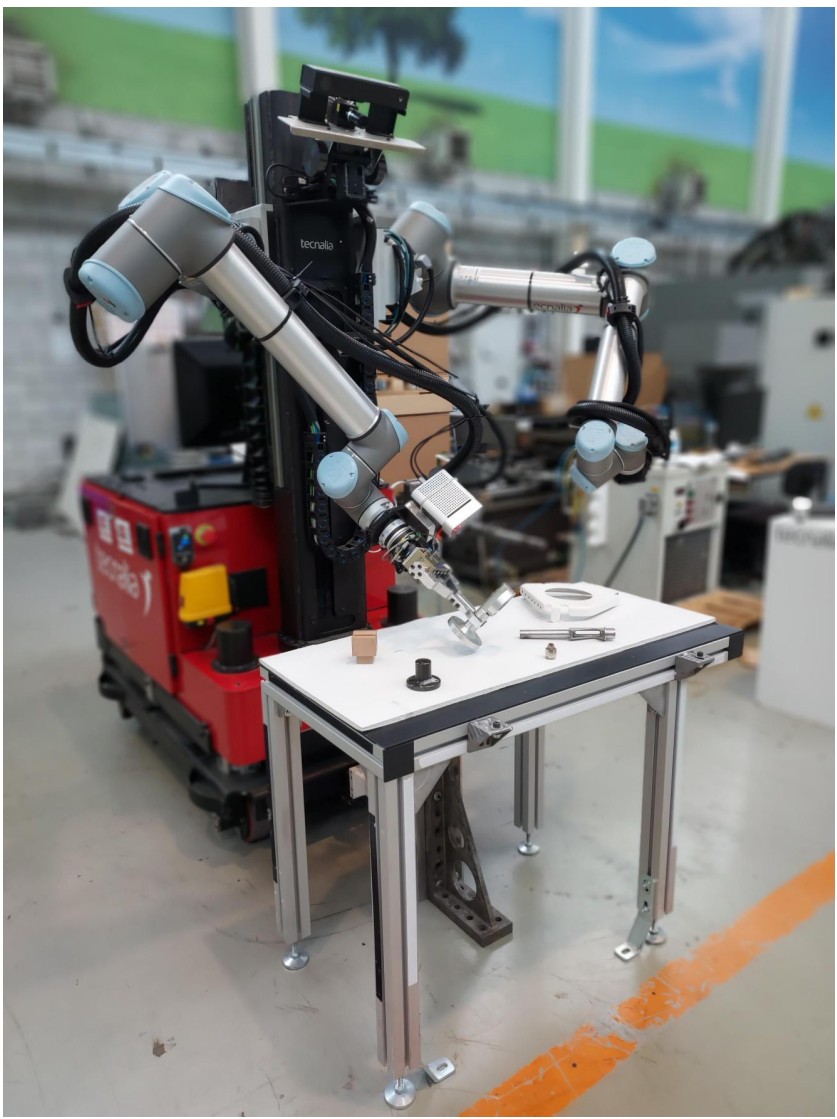

**Figure 1.** The AIMM used to carry out the tasks of the presented industrial application.

The robot packs many specially customized features that differentiate it from common commercial mobile manipulators.

- Large 200 A/h lithium battery for full shift operation. Its location was also designed to lower the robot's center of mass and help stability.
- Small capacity on-board pneumatic system enabling the use of pneumatic devices attached to the arms and a coupling mechanism to connect to an external compressed air system for higher demand applications.
- A docking mechanism that allows a physical connection between the robot and an external station for high accuracy positioning and resource exchange (opportunity charging, compressed air, input/output signals, etc.).
- A pneumatic tool exchanger system, allowing the robot to change the arm's tool on demand for different tasks. For the manipulation task presented in this paper, a customized pneumatic gripper Schunk DPG 100-1 is used (Figure 2).
- A wide range of sensors to improve its perception capabilities while complying with safety standards, including multiple encoders, ATI Delta torque sensors, two Sick S300 safety lasers, a Pixhawk 4 IMU and multiple optical (IDS uEye GigE) and RGB-D (Realsense D435, Kinect 2.0, Azure Kinect DK) cameras mounted on different points of the robot.

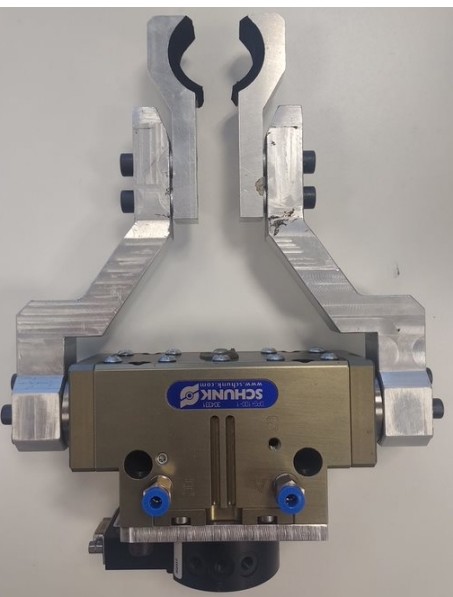

**Figure 2.** The pneumatic gripper used for grasping the detected objects.

### 2.1.1. Autonomous Navigation

One of the biggest advantages of AIMMs compared to traditional production line robots is their ability to change its workplace and to do it by its own means, navigating autonomously. This represents a great differentiating leap since there is no need to install a robot for each work cell, and a single robot can perform different tasks since its perception system allows it to adapt to each of them.

Two different levels of navigation were considered: the first one is a more general autonomous navigation in which the robot must navigate between targets, where low accuracy is required. This is typically required when moving along the workshop between workstations (global autonomous navigation). The second level is accurate autonomous navigation, in which the robot navigates in order to place itself very accurately against a target. This happens when the robot arrives to its workstation and needs to accurately place itself in the cell to successfully carry out its tasks (accurate docking).

#### Global Autonomous Navigation

The global navigation approach is the robot's ability to navigate between different work cells taking into account its environment. This type of navigation normally covers long distances and the accuracy requirements are in the range of several cm. The main objective is to safely and efficiently traverse the space between two approximate points.

Since 2D laser-based navigation is a well-established and proven technology [19], the approach followed has been to use out of the box components of the ROS navigation stack correctly parameterized for the kinematic characteristics of the AIMM. These components have been combined and augmented with 3D information from additional sensors to overcome some limitations of 2D navigation. Following the traditional approach, in a first learning phase Simultaneous Localization And Mapping (SLAM) [20,21] is used to generate a 2D occupancy map. The implementation used is the SLAM approach from [22], available in ROS as the package *hector_mapping*.

In the second phase, the previously recorded map is used to locate and generate traversable paths. Localization is based on Augmented Monte Carlo localization (AMCL) from [23], available as the popular *AMCL* ROS package [24].

Path planning is divided into global and local planning. Global planning geometrically computes a route between the robot's origin point and the destination point. In this proposal, the Dijkstra algorithm [25] is used, which is implemented in the ROS *global_planner* [26] package.

Local planning is responsible for computing the speed commands required to follow the route estimated by the global planner. It also modifies the original path to avoid collisions in case obstacles not present during global planning are detected. The ROS package used is *teb_local_planner* [27], which implements the Timed Elastic Band (TEB) approach [28].

Finally, the 2D navigation system was enhanced with the use of 3D information sources. The Intel RealSense D435 3D camera installed on the robot's torso provides a forward overview of the environment, detecting 3D obstacles that are projected into the costmap used for navigation.

A more detailed description of these systems and their implementation can be found in the previously mentioned report [15].

Accurate Docking

After the global autonomous navigation has driven the robot to the proximity of the operation zone, it is necessary to perform an accurate positioning to ensure a correct final placement. This accurate final placement is required to guarantee that the pieces to be detected fall within the field of view of the detecting camera.

To that end, a visual servoing-based docking system has been developed using Fiducial markers. The system is based on a proportional control that maintains and ensures, within a desired tolerance, the position of the robot with respect to the marker. A high-performance IDS Ueye camera with a sampling rate of 20 Hz is used to capture the images, allowing high control rates. Since the transformation between the camera and the robot's body is rigid, it is not necessary to perform an accurate calibration of the camera's position. Instead of directly using the estimated error in the image, the detected marker position is transformed to the robot's frame and compared with a previously recorded one from a calibrated position. This allows the camera to be mounted on any part of the robot, and in any orientation, without requiring accurate camera calibrations or modifications on how the speed commands are computed in the control program. The system is parameterizable to achieve specific speeds and accuracy. In Figure 3, the AIMM making a precise approach to the charging station can be seen.

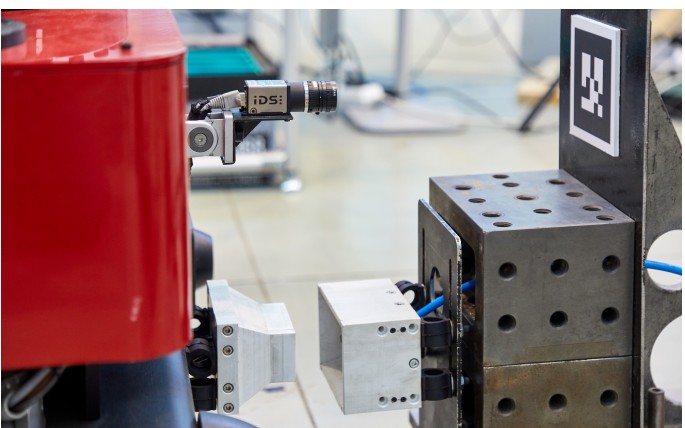

**Figure 3.** AIMM performing the accurate docking by detecting the Fiducial marker with the camera.

*2.2. Perception System*

In mobile manipulation applications, 3D object localization is very important because the relative position of the objects to work on is subject to uncertainty. Thus, the ability to recognize the object's 6D pose (i.e., to estimate object's 3D position and 3D orientation) is essential in order to grab such objects successfully.

Very common approaches are the methods based on the processing of RGB images to estimate the 6D pose of the objects. Most classical methods rely on detecting and matching keypoints with known object models [29–32]. While those approaches have good results,

they generally deal badly with texture-less objects, so it still remains a challenging problem. More recent approaches apply deep learning techniques for 6D pose estimation [33–35].

Recently, methods that use 3D point clouds from RGB-D images have been introduced [36,37]. While RGB-D methods are still not comparable in performance to RGB methods for pose estimation [38–41], those methods combine visual and geometrical features, making 3D predictions more accurate. Therefore, RGB-D methods fit better for our application because they obtain more accurate predictions for 6D pose estimation than RGB methods.

### 2.2.1. Used Method

Our system uses PVN3D [40] to estimate the pose with 6DoF. Figure 4 shows the overall architecture of the perception module which is composed of the PVN3D architecture and a synthetic dataset. The PVN3D works on two types of features. From the RGB images, it extracts appearance information using a Convolutional Neural Network (CNN) composed of a PSPNet [42] with ResNet34 [43] pre-trained on ImageNet [44]. From the depth images, it uses a PointNet++ [45] to extract geometric features. Both appearance and geometric features are fused by a DenseFusion block [39] and are passed to a three parallel modules block composed of a 3D keypoint detection module ($m_k$), a per-point semantic segmentation module ($m_s$) and a center offset voting module ($m_c$). Those three modules are composed of shared Multi-Layer Perceptrons (MLPs). With the resulting semantic segmentation and center voting modules, a clustering algorithm [46] distinguishes different instances and for each instance point it votes for their target keypoints. Finally, a least-squared fitting algorithm [47] is applied to obtain the 6D pose of each instance, in the form of the rotation $R$ and translation $t$ that transforms from the object coordinate system to the camera coordinate system. These are obtained by minimizing the loss from Equation (1). $\mathbb{I}$

$$L_{\text{least-squares}} = \sum_{j=1}^{M} ||kp_j - (Rkp_j' + t)||^2 \tag{1}$$

where $M$ is the number of selected keypoints, $kp_j$ are the detected keypoints in the camera coordinate system and $kp_j'$ are their corresponding keypoints in the object coordinate system.

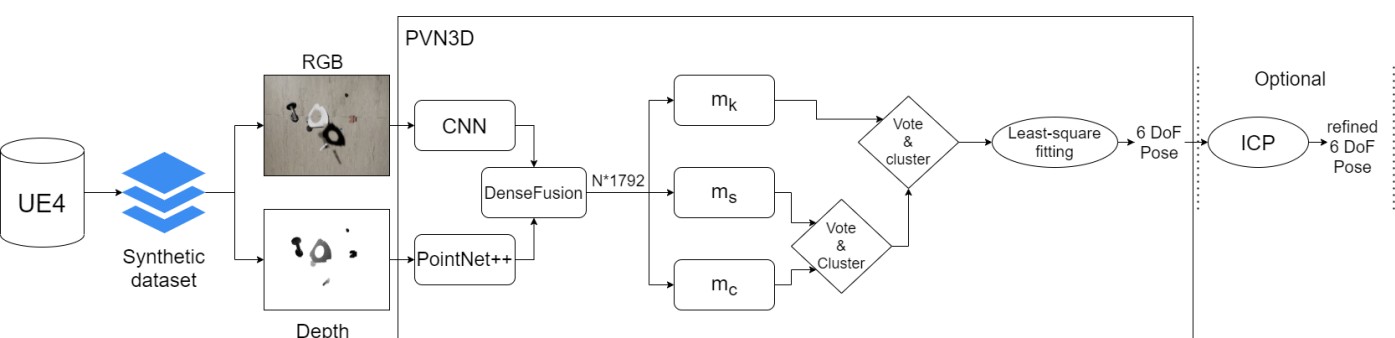

**Figure 4.** Perception module architecture.

### 2.2.2. Synthetic Dataset

A new training dataset has been created using synthetic images from the CAD models of the actual application's parts to be manipulated. As stated before, the PVN3D system has been trained using this synthetic dataset. Labeling 3D data is arduous and usually in industrial contexts it may be impossible to generate big datasets, or at the least very difficult, from real data. That is why, in many applications [48–51], synthetic data have been used recently with more frequency.

In our case, we use Unreal Engine 4 (UE4) to create the synthetic dataset. The Nvidia Deep learning Dataset Synthesizer (NDDS) [52] is a plugin of UE4 that enables to create high-quality synthetic images for deep learning applications.

The scene created is set up with a directional light, an atmospheric fog, a background plane, a distractor group, the training object group, an scene capturer and a scene manager:

- The directional light has 3.1415 lux of intensity and goes towards the background plane.
- The atmospheric fog makes the scene more realistic by smoothing far objects.
- The distractor group is composed of different geometrical shapes that are randomly generated and located over a greater area than the field of view (FoV) of the camera. That way, not every image shows all the distractors. These distractors help the deep learning methods to avoid overfitting. The number of distractors generated every 3 s is between 40 and 60.
- The background plane is randomly colored or given a texture. The color is picked from a palette of colors and the textures are random images from our laboratory's floor or workspace.
- The training group is a set of 6 objects to be detected. All of the objects are common industrial parts used in some of our projects. The spawn area is within the FoV of the camera. Thus, on every image all of the objects are present.
- The scene capturer is a virtual camera with a focal length of (320, 320) px, a principal point of (320, 240) px and a resolution of (640 × 480) px. This camera is in charge of capturing the features, including object data, color, depth, instance segmentation and class segmentation. The camera makes ten thousand captures.
- The scene manager is an optional component dedicated to control the segmentation options.

From each scene, a single RGB-D image is extracted and the scene is modified by randomly changing the 6D poses of the objects. The dataset is composed of ten thousand RGB-D images, with their corresponding semantic segmentation and instance segmentation images, and information of the objects on each image like 6D poses (locations + rotations), 3D bounding boxes and projected 2D bounding boxes.

Real cameras are not able to capture the reality with exact accuracy (i.e., they always present some kind of noise). Synthetic datasets do not present this variability unless it is specifically introduced to make the synthetic images more similar to real ones. Thus, some filters are applied to the synthetic images to that effect, both to RGB and depth images:

- The RGB images during training are filtered with the *rgbnoise*. This filter is applied only once with a probability of 0.8 and twice with a probability of 0.2. The filter is composed of an HSV augmentation, a linear motion blur with a probability of 0.2 and a Gaussian filter with a probability of 0.2. The Gaussian filter is applied with a window of 3 × 3 for 80% of the cases and with a window of 5 × 5, otherwise.
- The depth images are filtered during training with a Gaussian filter with a window of 3 × 3 in the depth component, keeping the spatial distribution.

Figure 5 shows the 6 real objects to be detected. All of them are industrial parts that have been used in our projects.

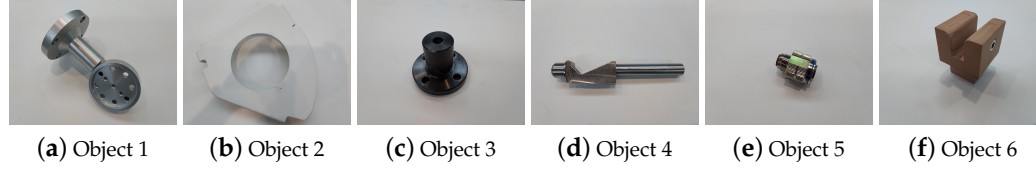

(**a**) Object 1    (**b**) Object 2    (**c**) Object 3    (**d**) Object 4    (**e**) Object 5    (**f**) Object 6

**Figure 5.** Set of six industrial objects selected to appear in our dataset.

Figure 6 shows an example of RGB, depth and semantic segmentation images obtained from the UE4. RGB and semantic segmentation images are 8-bit images. Depth image is a 16-bit image measured in cm.

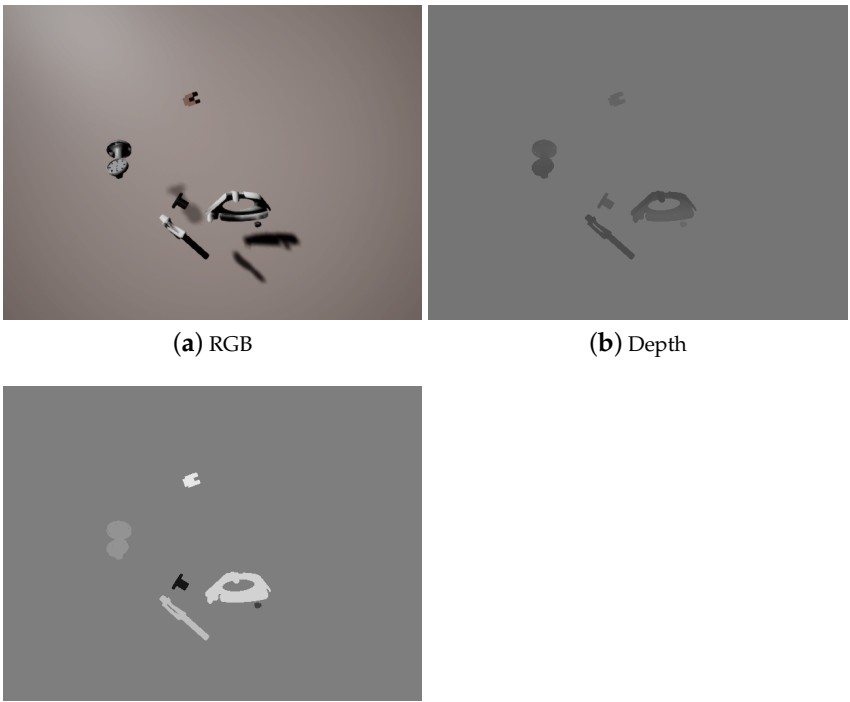

(**a**) RGB    (**b**) Depth

(**c**) Semantic segmentation

**Figure 6.** Output images from the UE4. Depth image has been adapted for visualization.

### 2.3. Calibration Process

An accurate manipulation system requires that all of the components that will intervene in the detection and manipulation processes are accurately calibrated. To calibrate the vision system, it is necessary to obtain the intrinsic and extrinsic parameters of the camera. The intrinsic parameters are the representation of the optical center and focal length of the camera, while the extrinsic parameters represent the camera's position relative to the world, in this case relative to the robot.

To calculate the transformation $^{flange}H_{camera}$ between the robot flange and the camera, which has been mounted in an eye-in-hand configuration, an automated hand-eye calibration procedure has been used. Based on an initial approximate camera pose $^{flange}H'_{camera}$ provided by the user and an initial calibration pattern detection $^{camera}H_{calpattern}$, an approximate pose of the calibration pattern is estimated as

$$^{base}H'_{calpattern} = {}^{base}H_{flange} * {}^{flange}H'_{camera} * {}^{camera}H_{calpattern} \qquad (2)$$

This approximate calibration pattern pose $^{base}H'_{calpattern}$ is used to calculate a set of robot poses $^{base}H^i_{flange}$ that will be used to move the robot and obtain new images where the calibration pattern can be found. Specifically, the poses are calculated as

$$^{base}H^i_{flange} = {}^{base}H'_{calpattern} * {}^{calpattern}H^i_{camera} * {}^{flange}H'^{-1}_{camera} , i = [1...n] \qquad (3)$$

where the approximate calibration pose $^{base}H'_{calpattern}$ and camera pose $^{flange}H'^{-1}_{camera}$ are provided in the previous steps and the relative poses between calibration pattern and camera $^{calpattern}H^i_{camera}$ are generated automatically, adding some small random translations and rotation.

Based on these poses, the robot is automatically moved to acquire images and estimate the calibration pattern pose in each of these images. This procedure will generate a dataset of *N* pairs of robot and calibration pattern poses.

$$data_i = (^{base}H^i_{flange}, \,^{camera}H^i_{cal\,pattern}). \tag{4}$$

These data are used to estimate the final camera pose $^{flange}H^*_{camera}$ using Ceres Solver [53] to perform a non-linear optimization where the cost function is calculated as:

$$cost_i = (^{base}H^i_{flange} * {}^{flange}H^*_{camera} * {}^{camera}H^i_{cal\,pattern})^{-1} * {}^{base}H^*_{cal\,pattern} \tag{5}$$

where the camera pose $^{flange}H^*_{camera}$ and calibration pattern pose $^{base}H^*_{cal\,pattern}$ are optimized during the process.

In addition, the calibration of the gripper installed in the flange of the robot is also necessary. The center of the two fingers has been calibrated as the tool center point (TCP) by means of the *three point method* provided by the robot manufacturer.

With these calibration processes, the entire system consisting of the arm, camera and tool (end-effector) has been referenced in a common frame.

### 2.4. Process Control and Programming

#### 2.4.1. Skill-Based Programming

Industrial tasks are usually composed of a sequence of operations, each dependent on the previous one. Thus, the AIMM has to be able to perform this ordered sequence while managing and controlling the execution flow and any possible errors. Following the advances presented in [54], where the concept of "skill" is presented as a nominative entity of a capacity learned by the robot, which, due to its similarity with human behaviors, make its understanding and handling easier.

In this work, the concept of skill is used to represent each capacity that the robot has acquired. Thus, it is easier to identify what skills are needed to perform a given task. For the use case presented in this paper, the most relevant skills are the following:

- Navigation: composed by the techniques presented in Section 2.1.1: Global Autonomous Navigation;
- Docking: wraps the capacity shown in Section 2.1.1: Accurate Docking;
- Object pose estimation: allows using the techniques presented in Section 2.2;
- Object grasping: successive robot movements that align the robotic arm to the detected object to, then, actuate a gripper that allows grasping an object;
- Undocking: using the same concept of docking but in the inverse way (where less accuracy is required).

To represent the execution flow, the concept of state machines is used. These machines manage the skills used to perform a certain task. This state machine approach facilitates programming for non-expert operators, making it possible to do modifications to the robot's tasks in a simple and visual way, including management of possible errors. In the following sub-section, more details about the implemented approach can be found.

#### 2.4.2. Skills Management by State Machine

Once all the robot skills necessary for the correct execution of the task have been defined, it is necessary to manage them. Each skill has an internal flow of atomic operations that provide an output. Moreover, a skill can suffer errors or not be able to perform its function correctly. To manage all possible cases, it is necessary to use a higher level tool, in a higher layer of abstraction. In this work, the use of state machines allows managing and orchestrating the developed skills required for each task. After testing different state machine libraries like [55] and custom developments, following works such as [56], it was decided to use Flexbe [57] as a manager.

Flexbe is a behavior engine that allows the generation of state machines to encode behaviors in robotic and automation applications. As can be seen in Figure 7, the skills can

be sequenced in a state machine that manages the required behavior, linking the output and inputs of the skills and handling the errors that eventually can occur.

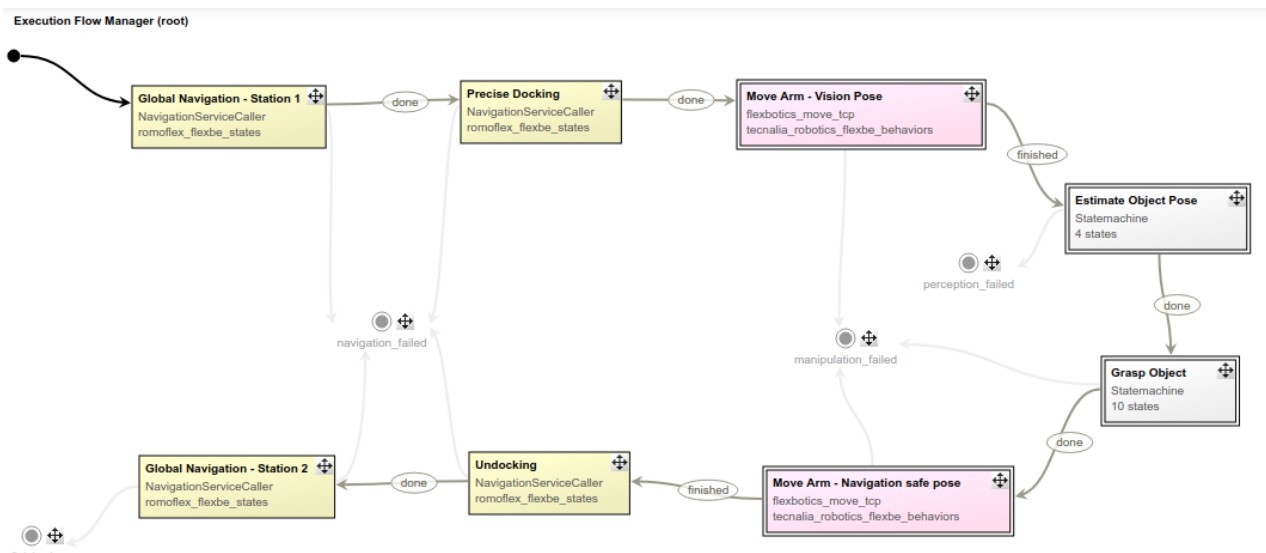

**Figure 7.** Developed skills sequenced in a Flexbe behavior.

The required functionalities to carry out these tasks have been encoded in a modular way organized in different ROS packages that offer an interface (common communication mechanisms in ROS, such as services [58] or actions [59]) that allow them to be invoked by the task manager. These blocks can be made up of different sub-blocks, conforming the previously mentioned skills (Section 2.4.1). This block can also have different outputs depending on the result. In the performed experiment, the object estimation skill is composed by a set of required operations that complement the pure vision operation (see Figure 8). First of all, the mobile elements must be stabilized and, after the perception result is obtained, it must be transformed to the robot base coordinate frame for further use.

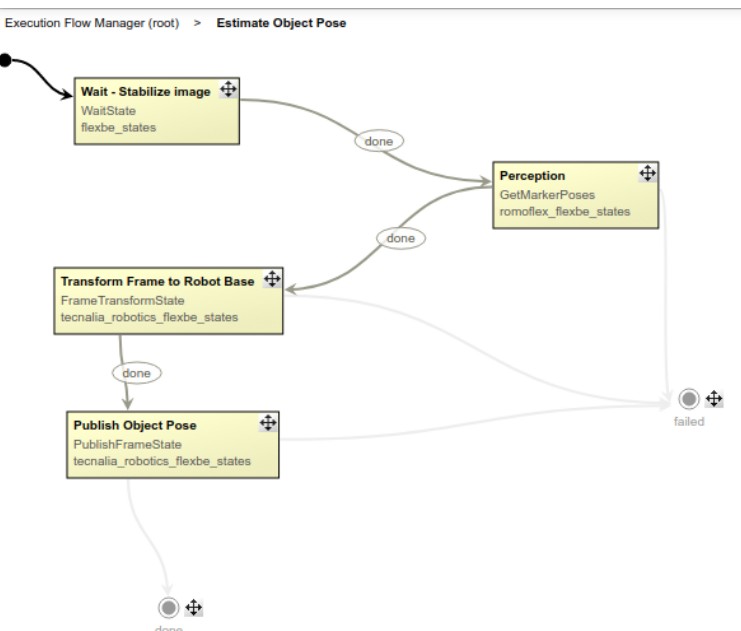

**Figure 8.** Estimate object pose skill is composed of additional operations that complement vision algorithms.

Regarding the grasp skill, taking as input the object estimated position computes the required approach, pre-grasp and retreat poses. The grasp skill sequences the calculated waypoints and interacts with the gripper for grasping the object when the robot arm is precisely aligned to it. Figure 9 illustrates all the involved states in the grasp operation.

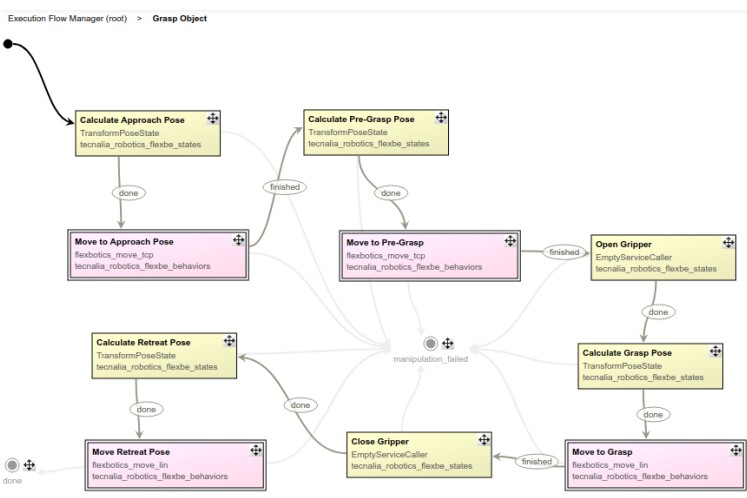

**Figure 9.** Grasp skill is composed of sequential robot movements and gripper operations.

The complete operation has been shown in Figure 7. There, the estimation and grasp skills are complemented with navigation and docking skills to complete the full operation.

## 3. Experiment Definition

The experiment consists of the correct execution of a paradigmatic operation for an AIMM in modern industry: to go to some workspace and manipulate a part. This operation is composed of a series of tasks which include autonomous navigation, artificial perception, manipulation and grasping and global management of the application execution flow.

The starting point is established in the electrical charging spot of the robot. A new task order is received in which the point goals where the robot has to navigate autonomously are established. By processing the data from its sensors, the robot is able to correctly locate itself in the environment and navigate towards the established objective. As mentioned in Section 2.1.1: Global Autonomous Navigation, the technology used, based on SLAM by a 2D laser, has an intrinsic error that can reach several centimeters of deviation in the location. Thus, a final precise navigation process is required once in the proximity of the goal to achieve a predefined and known required accuracy. This accurate position will ensure that the part falls both in the camera's field of view and the space reachable for grasping.

The robotic arm moves to a known position to focus the camera on the target object area. The camera used is an Azure Kinect DK by Microsoft. This camera is a developer kit that contains a best-in-class 1MP depth camera, 360° microphone array, 12MP RGB camera and orientation sensor. The RGB and depth images are obtained with the Azure Kinect DK and passed to the perception module. First, the point cloud is obtained from the depth image. That point cloud is filtered to clean noise, background data and the floor. The cleaned data with the RGB image are passed through the trained PVN3D model and the 6Dof Pose is predicted. The predicted pose is then refined with ICP to improve the results.

After detecting the desired object and obtaining its relative position in space, the grasping task is launched. In normal operation, this task would be composed of an approach maneuver by the robotic arm plus grabbing the object by means of the pneumatic gripper. However, developing or implementing an accurate grasping strategy is outside the scope of this paper. Thus, we will consider the grasping task successful if the gripper is able to touch the centroid of the object, in the assumption that, if given that reference point, a proper grasping method would be able to grab it correctly.

If there is no error and the object is correctly grasped, the arm is retracted to a safe navigation position that prevents it from colliding with obstacles. Then the AIMM navigates autonomously to the position where an operator will receive the requested object, completing the operation.

Possible errors in the application are managed through specific states of the Flexbe state machine. These states control the possible errors of the task execution process and take actions accordingly in each case.

## 4. Results

The results obtained can be measured by the success of both the whole operation and of each of the individual tasks it is composed of.

The results of the navigation and accurate docking part have been successful. In forty tests of navigation and accurate docking done in the real test environment, the operation was successful 100% of the times from different starting positions and with target positions in different orientation configurations.

To evaluate the maximum tolerances of the accurate docking, a battery of tests has been performed setting multiple starting locations with attack angles with respect to the docking station ranging in ±50°, at different distances. In this setup, the system was able to successfully dock from almost any distance, failing only at extreme angles (>45°) at very close distances (<60 cm), where even small movements can take the marker away from the camera's field of view. In all other cases, the robot is capable of carrying out the accurate docking process without any problem.

The perception module has been trained with the synthetic dataset, using a proportion of 0.85 for training and 0.15 for testing. The metric used to test the perception module is ADD(S) [36,38,39], as the authors of PVN3D [40] evaluated it. The results are shown in Table 1.

Overall, location and semantic segmentation are done correctly for all the objects, but rotation is not correctly obtained for some of them. Furthermore, if we compare the three losses of the three submodules of the perception module (Table 1), we can see that the semantic segmentation module and center offset module losses are of, at least, an order of magnitude less than the keypoint detection module ones. This is why, in many of our experiments, location and segmentation of the objects are predicted correctly but rotation is not that accurate. The symmetric objects are better detected than the not symmetric ones because there are more possible rotation solutions on symmetric objects. Even if rotation is not really accurate, this can sometimes be fixed by applying an ICP to the obtained pose. This refinement only improves the obtained pose if the initial pose is similar enough to the ground truth. All the detection phases are shown in Figure 10.

**Table 1.** Losses and evaluation of perception submodules on ADD(S) obtained in the test. Symmetric objects are in bold.

|  | $L_{keypoints}$ | $L_{semantic}$ | $L_{center}$ | $L_{multi-task}$ | **ADD(S)** |
|---|---|---|---|---|---|
| Object 1 | 0.9738 | 0.0275 | 0.0427 | 1.0715 | 5.12 |
| Object 2 | 1.8951 | 0.0278 | 0.0538 | 2.0045 | 1.02 |
| **Object 3** | 0.4216 | 0.0068 | 0.0223 | 0.4574 | 86.05 |
| Object 4 | 0.9789 | 0.0146 | 0.0451 | 1.0533 | 8.77 |
| **Object 5** | 0.1795 | 0.0077 | 0.0076 | 0.2025 | 68.89 |
| **Object 6** | 0.4247 | 0.0092 | 0.0157 | 0.4587 | 87.47 |

As can be seen in Figure 11, the tests carried out with real data have obtained a similar behavior. For almost all of the objects the position is predicted correctly, but the rotation incorrectly. Object 4 is incorrectly located in almost all the testing images. The possible cause could be that its metallic surface has many reflections that create lots of noise in the camera.

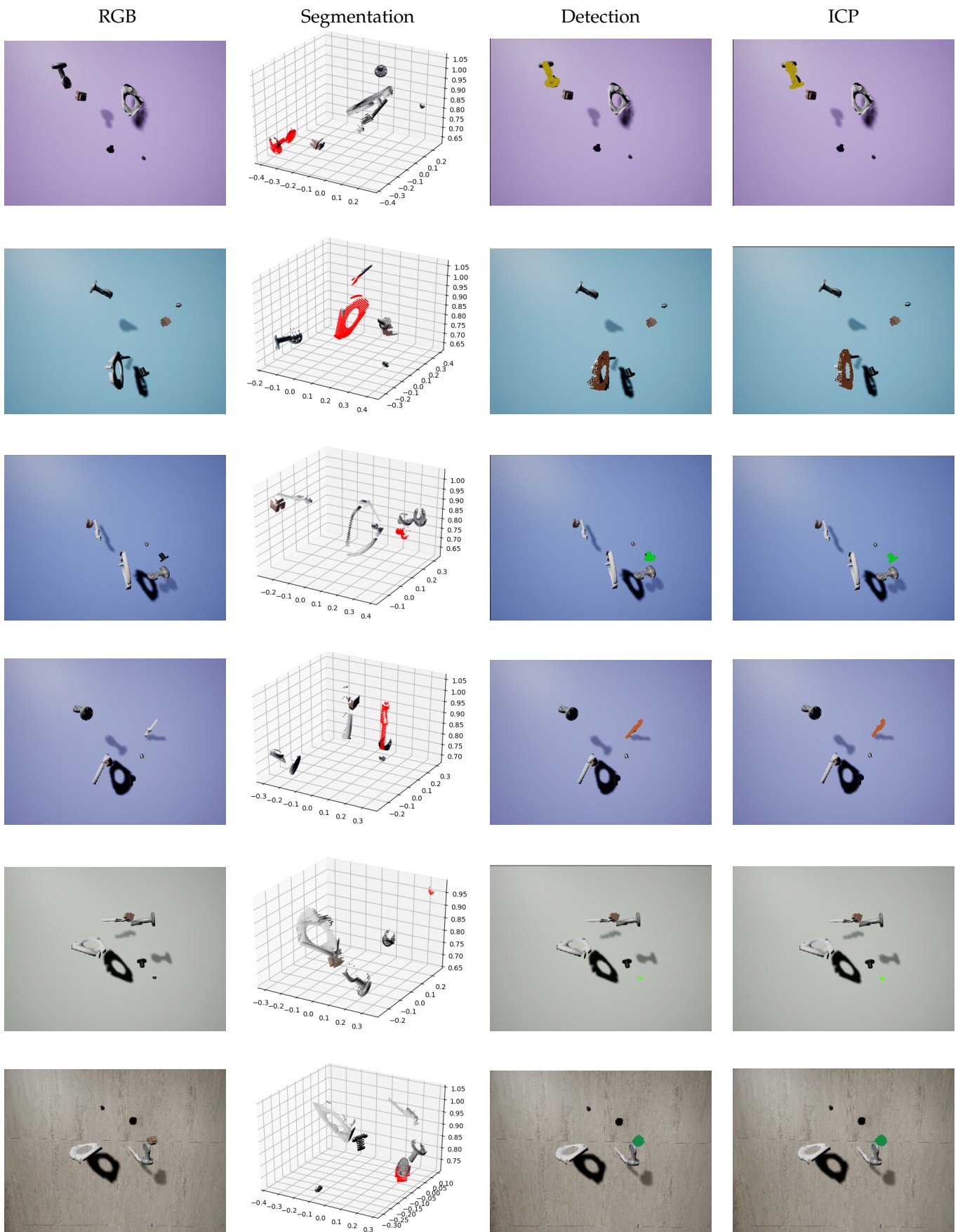

**Figure 10.** Detection results of the synthetic data. Each row corresponds to an object from Object 1 to Object 6.

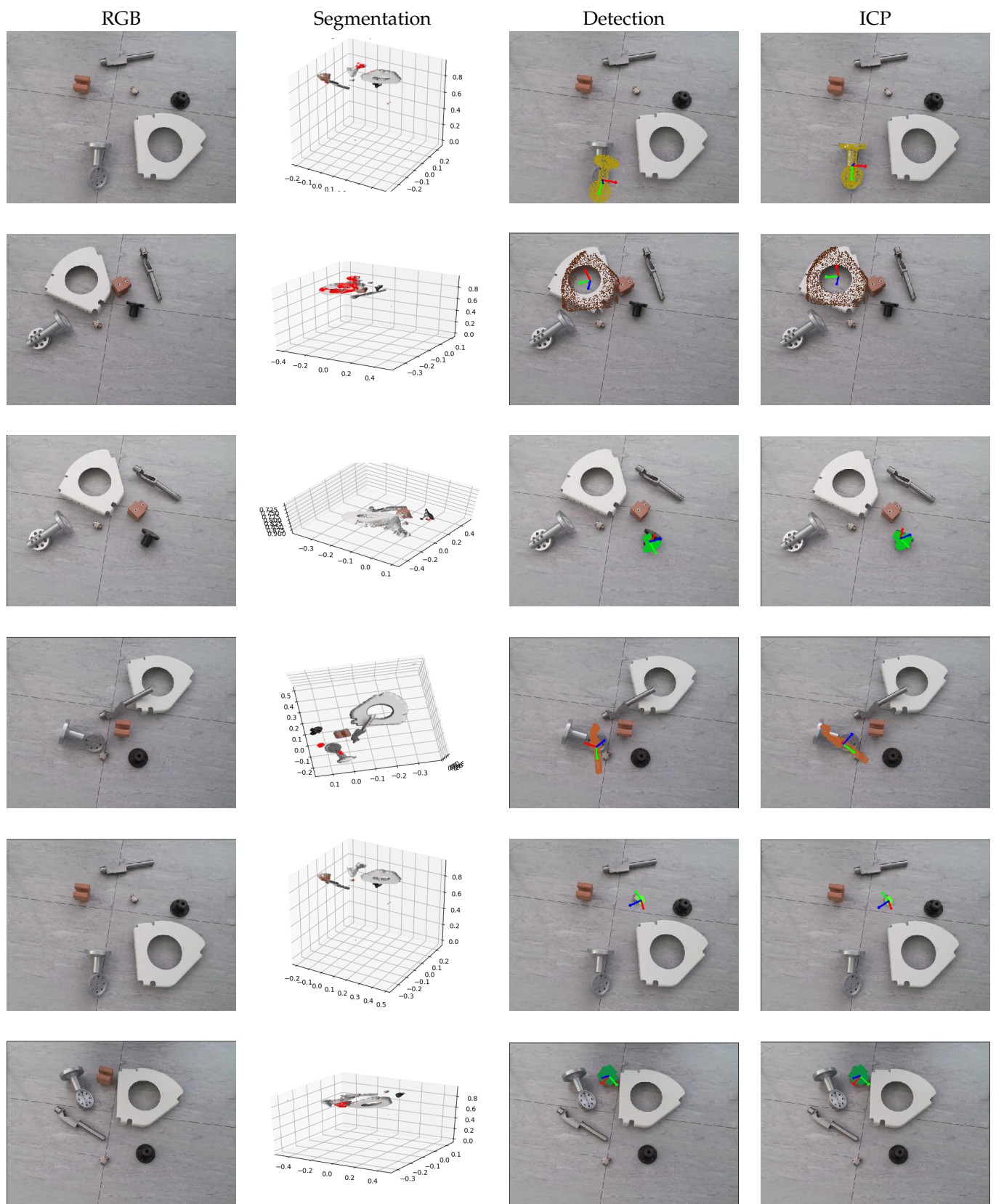

**Figure 11.** Detection results of the real objects. Each row corresponds to an object numbered from Object 1 to Object 6.

The camera–robot calibration process explained in Section 2.3 returns a calibration transformation given a set of images. For each image, the $^{base}H_{cal\,pattern}$ transformation can be calculated using the optimized transformation. In a perfect calibration, all the resulting poses should be equal although it is hardly possible as the process includes errors such as intrinsic robot calibration errors and camera calibration errors. Therefore, the mean $^{base}H_{cal\,pattern}$ transformation error (i.e., the average error in the estimation of the calibration pattern's position) in this setup is 0.00166856 m in translation and 0.00386598 rad in rotation. Based on our experience calibrating other robotic systems, these values are considered accurate enough to correctly perform the proposed grasping task based on the used hardware.

The overall operation has been done thirty times. The obtained results are shown in Table 2. The first column named "# executions" indicates the number of times the overall application has been performed. The second column "# successful executions" shows how many times the application has been executed successfully (that is, the AIMM has navigated autonomously to the target goal, performed the docking process accurately and finally detected and touched the centroid of the object). The achieved success rate is directly co-related with the output of the perception module. For instance, Object 4 is the object that obtains the worst results because the perception module predicts its pose incorrectly.

**Table 2.** Overall operation results.

|  | # Executions | # Successful Executions | % |
|---|---|---|---|
| Object 1 | 7 | 6 | 85.71 |
| Object 2 | 4 | 3 | 75 |
| Object 3 | 5 | 5 | 100 |
| Object 4 | 3 | 0 | 0 |
| Object 5 | 6 | 6 | 100 |
| Object 6 | 5 | 5 | 100 |
| Total | 30 | 25 | 83.33 |

## 5. Discussion

The main bottleneck in our application is the perception module. The problems reported in the estimation of the orientation part of the objects' 6D pose make the grasping operation difficult. Thus, the perception module must be improved before trying to introduce any grasping strategy in the system. The keypoints detection module is the most complex task to be carried out within the perception system. This is because the system needs to learn specific points of the objects in the scene. Thus, a dataset very rich in variability is critical to successfully train it. While the current dataset's variability seems enough for the other training tasks, it is possible that it is still lacking for training the keypoints. Further effort should be put into improving the dataset's variability.

One possible source of the problems detected in the perception module probably comes from the synthetic dataset. Due the complex nature of the RGB-D systems, the synthetic dataset may not be realistic enough to simulate a real one, especially regarding the noise in the depth information. While the introduction of synthetic noise in RGB is a well studied problem and can produce realistic results, the noise introduced in the depth information by reflections or different colors is a very complex problem for which no proper model exists yet to the knowledge of the authors. Thus, the dataset does not store properly the variability introduced by this type of noise. A way of introducing more realistic noise in the depth component should be studied further.

Another way of improving the dataset by increasing its variability would be to introduce more objects to help the model to learn more generic features. In addition, according to [60], combining domain randomization datasets, which is the type of dataset we generate, with photorealistic images increases the performance of models on synthetic datasets. All mentioned aspects considered, future works could include adding synthetic photorealistic datasets generated with Blender.

The developments presented in this paper have been carried out in a provisional AIMM while the robot for the execution of the use case of the European Sherlock project was being manufactured. After the validation of the real tests, current developments will be migrated and deployed in the new robot (Figure 12).

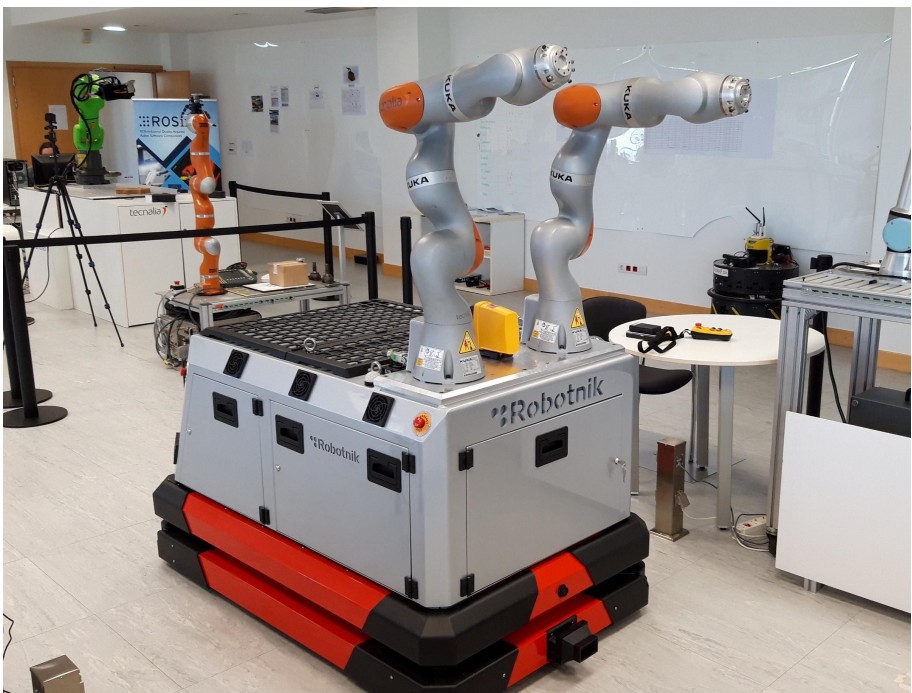

**Figure 12.** New AIMM of the European Sherlock project that was designed and manufactured during the testing and writing of this paper.

## 6. Conclusions

In this article, we presented a solution to a paradigmatic, usual task that AIMMs in industrial environments should be able to carry out, composed of navigation, perception and manipulation of objects.

To carry out this task, an AIMM equipped with the most modern sensors is used. Those sensors provide the robot with the capacity to perceive the environment in a wide range of forms (2D LiDAR, RGB, 3D), enabling it to achieve a high level of autonomy. A calibration process ensures that both the equipped sensors and the gripper used in the robot are accurately referenced.

Combining global navigation and accurate docking technologies, the robot is capable of autonomously navigating between different locations in the workspace and reaches targets accurately with defined tolerances. To detect the objects to manipulate, a deep-learning perception system based on 3D vision is used. Using the images obtained by the camera and a depth learning system, the robot is able to estimate the position of the object after having previously trained with synthetic data that represent real objects. The perception method is able to estimate the object's pose with high success rate, while having problems to estimate the orientation.

For the process' work flow management, a skill-based state machine system is used. Each operation to be carried out is decomposed in a sequence of smaller, atomic tasks that try to be recognizable and easily assimilable for human operators. These task or skills can be combined to create different sequences for different operations. The skill work flow for the operation is introduced in a state machine, where all the skills necessary to execute the task are ordered, monitored and managed.

The complete system presented, implemented in the AIMM, has been able to perform the proposed paradigmatic industrial operation in a real environment with an average 83.33% success rate.

The main contributions of this article are summarized in the correct achievement of a real industrial application through the use of an advanced robot and in the integration of different available techniques and methods in one system. Additionally, a new RGB-D training dataset that uses synthetic images of real industrial parts has been compiled and will be made available.

Further lines of work will include changing the synthetic dataset generation by adding photorealistic images with blender in order to improve the vision system, include a grasping module to manipulate the detected objects and migrate all the developed modules to the new robot. Within the framework of autonomous navigation, it has been proposed to add 3D information through the inclusion of new sensors (Lidar 3D, RGBD/stereo camera) to increase the perception of the robot and make it capable of navigating more robustly and safely using SLAM3D techniques.

**Author Contributions:** Conceptualization, J.L.O., I.M. and I.V.; methodology, J.L.O., I.M., P.D., A.I. and H.H.; software, J.L.O., I.M., A.I., P.D. and H.H.; validation, B.S. and I.V.; formal analysis, J.L.O., I.M. and I.V.; investigation, J.L.O., I.M., H.H. and A.I.; writing—original draft preparation, J.L.O. and I.M.; writing—review and editing, I.V. and B.S. All authors have read and agreed to the published version of the manuscript.

**Funding:** This research was funded by EC research project "SHERLOCK—Seamless and safe human-centered robotic applications for novel collaborative workplace". Grant number: 820683 (https://www.sherlock-project.eu accessed on 12 March 2021).

**Conflicts of Interest:** The authors declare no conflict of interest.

## Abbreviations

The following abbreviations are used in this manuscript:

| | |
|---|---|
| AIMM | Autonomous Industrial Mobile Manipulator |
| AMCL | Augmented Monte Carlo localization |
| CNN | Convolutional Neural Network |
| CPPS | Cyber-physical production systems |
| HRC | Human Robot Collaboration |
| HSV | Hue, Saturation, Value |
| MLPs | Multi-Layer Perceptrons |
| NDDS | Nvidia Deep learning Dataset Synthesizer |
| RGB | Red, Green, Blue |
| RGB-D | Red, Green, Blue, Depth |
| ROS | Robot Operating System |
| SLAM | Simultaneous Localization And Mapping |
| TEB | Timed Elastic Band |
| TCP | Tool Center Point |
| UE4 | Unreal Engine 4 |

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
