# Peer review of "A Real Application of an Autonomous Industrial Mobile Manipulator within Industrial Context"

_electronics, doi:10.3390/electronics10111276_

Round 1

Reviewer 1 Report

The topic of the article is very current and in line with the global trend in the field of production automation, autonomous vehicles and Industry 4.0 functions. Literature analysis has been done well but there is no reference to Industry 4.0 and the Internet of Things. The bibliography may be supplemented with several items in these areas.
The research object was accurately and correctly described. The experiment was planned and performed correctly. The research results are presented in an interesting way. The diagrams and drawings were made in sufficiently good resolution.
The discussion of the obtained test results was carried out correctly. The authors indicate that the research will continued. They also present another version of the autonomous robot.
The article is very interesting and can be of great practical use. Commercialization of the results of research and development works is very important.

Author Response

Thank you for your comments and for bringing to our attention the lack of references to the Industry 4.0. As a matter of fact, one of the key aspects of the EU project where this work is framed revolves around that concept. Thus, we have modified the introduction section to frame both this specific work and the project in general within the Industry 4.0 framework.

Reviewer 2 Report

The authors have tried to bring a serious work with this paper. There are a few imperfections those must be improved:

-   Look at the introduction and justification for the study in particular by using the literature to provide research gap. The contribution of this work should be clearer. This will situate the study in a better context.

-   The authors only give the results without a corresponding explanation. It is better to do some data analysis to make the conclusion more believable. You can improve the discussions by discussing the implications of each of the key findings. Thus, both theoretical/academic and practical implications. At present, this is weak in the paper.

-       There should be presented some recommendations in the conclusion as well as future study proposals. Also, conclusions should be rewritten to understand the importance of research. Are there any further research streams?

Author Response

The authors have tried to bring a serious work with this paper. There are a few imperfections those must be improved:

    • Answer: Thank you for your comments.

Look at the introduction and justification for the study in particular by using the literature to provide research gap. The contribution of this work should be clearer. This will situate the study in a better context.

    • Answer: The introduction has been modified to try to make more clear what are the contributions of the work. The project’s objectives have been updated and the Section 1.1. - Objectives expanded, listing the specific contributions. Namely, the successful implementation of the industrial task by the robot, integrating and tuning different available techniques and methods in one system and bringing them to a real world application. Also, a new RGB-D training data-set using synthetic images from real industrial parts has been compiled and made available.

The authors only give the results without a corresponding explanation. It is better to do some data analysis to make the conclusion more believable. You can improve the discussions by discussing the implications of each of the key findings. Thus, both theoretical/academic and practical implications. At present, this is weak in the paper.

    • Answer: As suggested, we have extended the Results and Discussion sections, adding further analysis of the results obtained both in the local results and in the overall result of the application.

There should be presented some recommendations in the conclusion as well as future study proposals. Also, conclusions should be rewritten to understand the importance of research. Are there any further research streams?

    • Answer: The Conclusions section has been extended with more details about future research lines for improving the achieved results and the contributions of the current work.

Reviewer 3 Report

In this paper, the authors presented their work on autonomous navigation of a mobile robot, and pose estimation to locate and manipulate different objects. While the project itself is interesting, there is not much technical or theoretical contribution. For e.g., for autonomous navigation, SLAM algorithm from ROS is used directly. Path planning was done using Dijkstra algorithm, also available from ROS. Visual servoing is done using standard methods. Object classification is achieved using ResNet and PointNet++. Using synthetic dataset for training is also not a new concept.

As such, I would not recommend the paper to be accepted for publication in this journal.

Author Response

Thank you very much for your comments. Indeed, the novelty of the work doesn’t reside in the individual methods used, but in the application as a whole and in the fact that we are solving a real world industrial problem. We consider that building up a complex system, correctly integrating and parameterizing several different and recent technologies in a novel robotic system for a real application in modern industry is a clear contribution. Also, a new training data-set composed of synthetic images of real industrial parts has been created from scratch and will be made available. We have updated the Introduction and the Conclusions sections so the contributions are more clearly presented.

Round 2

Reviewer 3 Report

The authors have highlighted the paper's contributions, and as such, have addressed my earlier concerns. The paper itself is well-written and it's good to see practical implementation of mobile robotic systems. As such, I recommend the paper to be accepted for publication.